# Impact of Biochar Addition and Air-Flow Rate on Ammonia and Carbon Dioxide Concentration in the Emitted Gases from Aerobic Biostabilization of Waste

**DOI:** 10.3390/ma15051771

**Published:** 2022-02-26

**Authors:** Mateusz Malinowski, Stanisław Famielec

**Affiliations:** Department of Bioprocesses Engineering, Energetics and Automatization, Faculty of Production and Power Engineering, University of Agriculture in Krakow, ul. Balicka 116b, 30-149 Kraków, Poland; stanislaw.famielec@urk.edu.pl

**Keywords:** biochar, municipal solid waste, undersized fraction, ammonia emission, carbon dioxide, aerobic biostabilization

## Abstract

Application of additives to waste may influence the course of the biostabilization process and contribute to its higher effectiveness, as well as to a reduction in greenhouse gas and ammonia (NH_3_) emission from this process. This paper presents research on the impact of biochar addition on the course of the biostabilization process of an undersized fraction from municipal solid waste (UFMSW) in terms of temperature changes, CO_2_ concentration in the exhaust gases, NH_3_ emission from the process, as well as changes in the carbon and nitrogen content in the processed waste. Six different biochar additives and three different air-flow rates were investigated for 21 days. It was found that biochar addition contributes to extending the thermophilic phase duration (observed in the case of the addition of 3% and 5% of biochar). The concentration of CO_2_ in exhaust gases was closely related to the course of temperature changes. The highest concentration of CO_2_ in the process gases (approx. 18–19%) was recorded for the addition of 10% and 20% of biochar at the lowest air-flow rate applied. It was found that the addition of 3% or a higher amount of biochar reduces nitrogen losses in the processed UFMSW and reduces NH_3_ emission by over 90% compared to the control.

## 1. Introduction

Aerobic decomposition processes are widely applied in facilities managing different types of waste rich in organic matter. These processes include composting (aimed at transformation of feedstock into organic fertilizer, i.e., compost) and biostabilization, whose main goal is to turn the processed waste into a stabilized material (stabilized waste), which can be safely landfilled or used in land reclamation. Biostabilization, also referred to as biological stabilization, is an aerobic decomposition process and is most often applied in mechanical biological treatment (MBT) in plants managing mixed municipal solid waste (MSW) [1,2]. The undersized fraction of MSW (UFMSW) obtained after screening (opening sizes ranging from 80 to 120 mm) is typically subjected to aerated bioreactors, in which an intensive phase of biological stabilization takes place (at least 2 weeks) [3,4,5]. Next, the waste undergoes a maturation phase, which usually is carried out in an open-windrow system. The whole proceeding, if properly conducted, leads to a reduction in the volume and microbial activity of the processed waste [6,7]. However, aerobic biostabilization of waste results in the release of carbon dioxide (CO_2_) and many deleterious gases into the atmosphere [8,9], and hence efforts are being made to minimize the negative impact of biological treatment methods and MBT facilities in this respect.

Gaseous emission from composting/biostabilization results from aerobic microbial activity in the treated waste, which is strongly determined by waste type and properties (e.g., carbon (C) and nitrogen (N) content, C-to-N ratio (C/N), dry matter (DM), and moisture content (MC)), as well as technological parameters of the process e.g., aeration and temperature [9].

Jędrczak [10] stated that the optimal C/N ratio for biological waste treatment is 25–35. For values of C/N ratio higher than 50, the decomposition processes occur at a slow pace, whereas in the case of a C/N ratio lower than 25, excess nitrogen may be released into the atmosphere, causing ammonia (NH_3_) emission in amounts that are toxic to the microbial population. According to Bilitewski et al. [11], the toxic effect of ammonium nitrogen is revealed at a C/N ratio lower than 15.

Another important factor that determines the effectiveness of a biological treatment process is temperature [12]. It affects the metabolism and population numbers of microorganisms, with a direct effect on gaseous emissions [9]. The highest increase in temperature (up to 60–70 °C) is typically observed in the first days of the process, and it is when the microbial activity, as well as the gaseous emission, is the most intensive. The period when the temperature exceeds 45 °C (the so-called thermophilic temperatures) can last for a couple of days and contributes to partial sanitation of the processed waste [2]. Intensive decomposition processes also result in increased CO_2_ emission, since it is a major product of biochemical oxidation reactions occurring in the treated waste [13]. Therefore, measurements of CO_2_ concentration in the exhaust gas are useful to evaluate the decomposition rate and assess the efficiency of waste treatment [12].

Gaseous emission during composting/biostabilization include, among others, volatile organic compounds (VOCs), NH_3_, carbon monoxide (CO), nitric oxide (NO), nitrogen dioxide (NO_2_), sulfur dioxide (SO_2_), hydrogen sulfide (H_2_S), and methane (CH_4_) [14]. NH_3_ is the most abundant alkaline gas in the atmosphere, and the major component of total reactive nitrogen [15]. Since it is toxic and odorous, NH_3_ contributes to malodor and health problems [16]. It plays a significant role in the formation of atmospheric particulate matter and atmospheric deposition of nitrogen into sensitive ecosystems [15]. Moreover, in the atmosphere, NH_3_ may undergo oxidation, producing nitrous oxide (N_2_O) [17], a greenhouse gas (GHG) with a global warming potential 265 times higher than that of CO_2_ [18].

During aerobic processes of organic matter decomposition (such as biostabilization or composting), organic N is mineralized and subsequently released as soluble NH_4_^+^ ions. As Stegenta et al. [9] stated, there are different competing pathways responsible for removing NH_4_^+^ ions from the liquid phase in the processed feedstock: (1) volatilization loss as NH_3_, (2) adsorption of NH_4_^+^ on organic and mineral surfaces, (3) immobilization by microorganisms, (4) nitrification to NO_3_^−^ (possibly followed by denitrification to NO_2_^−^ and N_2_), and (5) leaching and runoff. The nitrogen balance during aerobic treatment is controlled by complex biotic and abiotic interactions between waste matrix; microorganisms; and conditions influencing mass and heat transport, activity of microorganisms, and chemical and physical processes [9,12].

In the case of composting process, NH_3_ emission has been investigated in numerous research works. De Guardia et al. [19], who composted wastewater treatment sludge with woodchips, reported the accumulated emission rate within 20 days of the process varying from approx. 0.5 to as much as approx. 13.0 g NH_3_ per kg of initial DM (DM_init_), whereas Eklind et al. [12], who investigated composting of household waste mixed with straw, showed that the accumulated emission may reach 5.1 g NH_3_·(kg DM_init_)^−1^ within 15 days of the process. Pagans et al. [20] noticed that NH_3_ emission during composting strongly depends on the process temperature, with significantly higher NH_3_ concentrations in thermophilic temperatures. Similarly, Eklind et al. [12] showed that the emissions of NH_3_ increase with temperature, especially above 55 °C. Hanajima et al. [21] stated that NH_3_ emission occurs mainly in the thermophilic phase due to the strong biodegradation of organic nitrogen to inorganic nitrogen. As De Guardia et al. [19] reported, the aeration rate also influences the gaseous emission. They concluded that an increase in aeration (higher oxygen supply) is responsible for the increase in NH_3_ emissions. Similarly, Beck-Friis et al. [22], who composted mixtures of wheat straw and source-separated household organics, reported that the amount of NH_3_ emitted is higher with an increase in the aeration rate. Jiang et al. [14] found that while reducing CH_4_ emission, high aeration rates increased the amounts of NH_3_ and N_2_O emitted from the composting process.

A key factor in reducing GHG emissions and odors from the biological treatment of waste is properly set aeration. It contributes to maintaining optimal biological activity, and it is also a critical parameter affecting the emission of such gases as CO_2_, CH_4_, H_2_S, N_2_O, NH_3_, and VOCs [23]. Minimizing the gaseous emissions from biological processing can also be achieved by supplementing the input waste with natural structural materials [24,25,26] or digestate [2]. In the case of the comporting process, biochar has been reported as an effective additive [27,28,29].

Biochar is a carbon-rich material, produced mostly by the thermal treatment of biomass in the absence of oxygen. Typically, biochar is characterized by a high specific surface area and a water retention capacity [30]. Biochar can be produced from various substrates under different conditions [31].

The positive effects of adding biochar during composting of biowaste include a decrease in element losses and lower emissions of GHG, NH_3_, and odors [30,32].

There are two ways in which biochar contributes to lowering NH_3_ emission in the process gas during the aerobic biological treatment of organic matter [33]. Firstly, the development of negatively charged functional groups on the biochar surface is observed, which enhances the adsorption of cations such as NH_4_^+^ [34]. Secondly, biochar may adsorb organic N compounds, decreasing their mineralization [35,36] and consequently NH_3_ emission. A reduction in NH_3_ emission was widely observed in studies concerning biochar application to soils. Clough et al. [37] and Bai et al. [38] have reported that biochar adsorption of NH_3_ decreases NH_3_ and NO_3_^−^ losses from soil and offers a mechanism for developing slow-release fertilizers. However, the intensity of this process is affected by the interactions between biochar and environmental factors, the pyrolysis temperature of biochar, and biochar surface properties [39].

While there are many proofs that biochar is a good additive to composting [33,40,41], to the best of our knowledge, the impact of adding biochar on the biostabilization process and its gaseous emission has not been analyzed.

We hypothesized that biochar obtained from woodchips, as well as various air-flow rates during the aerobic process, would have a significant effect on the composition of the exhaust gases during the intensive phase of the UFMSW biostabilization process and on the accumulated gaseous NH_3_ emission. Moreover, this effect would depend on the amount of biochar added.

The main goal of this work was to evaluate whether, and if so to what extent, the application of biochar in six different doses and under various air-flow rates influences the temperature course of the biostabilization process (intensive phase), concentrations of CO_2_ and NH_3_ in the exhaust gases, as well as accumulated gaseous NH_3_ emission. Moreover, an attempt was made to correlate the measured gaseous concentrations and emissions with changes in the content of C and N, the C/N ratio, and the ash content for different doses of biochar additives and different air-flow rates.

The novelty of this research lies in describing the impact of biochar addition and air-flow rate on the biostabilization process (3-week intensive phase) of UFMSW separated from MSW. There is also a practical aspect of the conducted research, namely it contributes to the discussion on how to improve the biological treatment of waste at full-scale facilities and how to mitigate their negative effects on the environment, such as gaseous NH_3_ emission.

## 2. Materials and Methods

### 2.1. Materials and Process

The waste samples under study consisted of UFMSW exiting the mechanical treatment process (MSW treatment with a rotary screen, 80 mm opening size) at an MBT facility (MIKI Recykling Ltd.) in Kraków (Poland). The biochar used in the research is a material available commercially on the market and was obtained from coniferous woodchips as a result of pyrolysis in the thermalization energy recovery module at 550 °C for 30 min [29,42]. As Lehmann [43] and Jin et al. [44] have reported, this temperature allows for obtaining biochar with the most favorable properties, i.e., high carbon content, cation exchange capacity, and a specific surface area, which are important in the biological methods of waste treatment and biochar application to soils [45,46,47,48].

The tests were carried out using BKB 100 laboratory bioreactors (ROTAMETR, Gliwice, Poland), characterized by a 116 dm^3^ working volume and a 99 cm working height. Their construction and operation have been presented by Malinowski et al. [2] and Baran et al. [49]. The analysis covered the intensive phase of the aerobic biostabilization of UFMSW. The time of analysis in each run covered the first three weeks of the process. The initial temperature of the input mixtures was 14.4 ± 2.1 °C [42]. Information about the test conditions, the initial mass of waste placed in the bioreactor, and the methods of mixing the biochar with UFMSW were provided by Malinowski [42].

Six different doses of biochar obtained from coniferous woodchips were added to UFMSW, B1.5%, B3%, B5%, B10%, B20%, and B0%, without the addition of biochar as a control sample (% refers to the wet weight basis). The size of the applied biochar particles was <10 mm.

The biostabilization processes were repeated at 3 different average air-flow rates (0.1, 0.2, and 0.4 m^3^·d^−1^ (kg organic DM)^−^^1^, referred to further as 0.1, 0.2, and 0.4, respectively) for each biochar dose. The aeration intensity was controlled depending on the waste temperature in the bioreactors using the automatic control system equipped with a temperature feedback regulator. The controlling system reacted to prevent the temperatures in the bioreactors from exceeding 65 °C [42]. The applied air-flow rates were chosen based on the results of Neugebauer et al. [50], Yuan et al. [51], and Tom et al. [52].

### 2.2. Laboratory Tests

During the tests, the following parameters of the input mixtures were determined: DM, ash content, C and N content, C/N ratio, and N losses. Representative samples of the input mixtures were taken in triplicates at the beginning of the process (results marked as “initial”) and after 21 days and subjected to analytical tests.

The DM of raw materials, mixtures, and waste during and after the process was determined by the oven drying method. About 150 g of the sample was weighed with an accuracy of 0.1 g and evenly distributed on a tray. The sample was placed in a laboratory oven heated to 105 ± 2 °C. After 60 min, the tray with the sample was weighed. The process was repeated until the difference in the subsequent sample masses did not exceed 0.2% of the first weighing mass.

For each replicate, the percentage change in water content (in relation to the initial MC) was calculated. This value is one of the most important parameters indicating the effectiveness of the biological stabilization process. According to Jędrczak [10], MC is not a factor limiting the biological treatment of waste but its excessive loss (water loss) may lead to the drying of waste and a reduction in microbial activity. The ash content was determined by placing the sample in a muffle furnace at 550 ± 10 °C for 4 h.

The N content was determined using an ELTRA N 580 analyzer (ELTRA, Haan, Germany) in an oven preheated to 950 °C. The C content was determined using an ELTRA CHS 580 analyzer (ELTRA, Haan, Germany) combined with a furnace preheated to 1350 °C, in which the sample was burned and the flue gas was directed to the measuring cuvettes.

The N losses (N_loss_) were calculated from the initial (X_1_) and final (X_2_) ash contents according to the equations by Paredes et al. [53]:$${\;N}_{\mathrm{loss}}\left(\%\right)=100-100\frac{X_{1}\cdot N_{2}}{X_{2}\cdot N_{1}}$$
where N_1_ and N_2_ are the initial and final N concentrations, respectively.

### 2.3. Temperature and Gas Monitoring

The temperature was measured using Pt1000 probes placed in processed waste (inside the bioreactor). Analysis of the process gas was carried out using a NANOSENS DP-28-MAP analyzer (NANOSENS, Tarnowo Podgórne, Poland), which enables measuring O_2_ (±0.1%), CO_2_ (±1%), H_2_S (±1 ppm), NH_3_ (±1 ppm), and CH_4_ (±1%) content. The authors of this article focused on the effect of biochar addition and various air-flow rates on CO_2_ and NH_3_ concentrations in the process gases that were directed into the biofilter. Changes in oxygen concentration, bulk density, and air-filled porosity are described by Malinowski [42]. The concentration of individual gases was measured using two probes: one placed in the upper part of the waste bed and the other in the conduit directing the gases out of the bioreactor to the biofilter. The probes were periodically connected to the analyzer so that the gas composition could be measured.

The accumulated NH_3_ emission ($E_{NH_{3}}$) was evaluated according to the equation resulting from the relationships presented by Puyuelo et al. [23]:$$E_{NH_{3}}=\frac{1}{10^{6}}\cdot \frac{c_{NH_{3}}}{M_{TS}}\cdot \frac{p\cdot M_{NH_{3}}}{R\cdot T}\cdot F\cdot t$$
where $E_{NH_{3}}$ is the accumulated NH_3_ emission per kg of initial DM (g NH_3_·(kg DM_init_)^−1^), $c_{NH_{3}}$ is the NH_3_ concentration determined experimentally (ppm), $M_{TS}$ is the total initial DM of waste treated (kg DM_init_), *p* is the pressure (Pa), $M_{NH_{3}}$ is the NH_3_ molecular weight (g·mol^−1^), *R* is the gas constant (m^3^⋅Pa⋅K^−1^⋅mol^−1^), *T* is the temperature (K), *F* is the gas flow (m_3_·s^−1^), and *t* is time (s).

### 2.4. Statistical Analysis

Analytical tests of the raw materials and mixtures were run in triplicates. Statistical analysis of the obtained results was made using Statistica 13 software (TIBCO Software Inc., Palo Alto, CA, USA (2017). Statistica data analysis software system, version 13). An analysis of variance was performed in order to check the significance of the diversity of selected physicochemical properties in the samples obtained at various stages of the intensive phase during the biostabilization process of UFMSW with biochar.

## 3. Results and Discussion

### 3.1. Characteristic of Raw Materials and Mixtures

The average initial MC of UFMSW was 43.2 ± 0.9 wt%, while the OM measured by loss on ignition was 47.9 ± 0.8% DM, which indicates that the basic conditions for the biostabilization process were met [10]. The bulk density for UFMSW amounted to 459.5 ± 39.0 kg·m^−3^, whereas pH was 6.4 ± 0.1. Despite the heterogeneous material composition of UFMSW, the samples collected for analyses revealed physicochemical properties similar to those of the waste used in the research described by Malinowski et al. [2]. The average content of biodegradable waste in UFMSW was 43.4 ± 1.9 wt%.

The biochar used in this research was characterized by C content exceeding 80%; high air-filled porosity (AFP), reaching 85%; slightly acidic pH (6.0 ± 0.1); and low values of MC (4.5 ± 0.3 wt%) and bulk density (219.6 ± 4.8 kg·m^−3^). Regarding the above-mentioned parameters, they were similar to those of the biochar additive used in the research presented by Malinowski et al. [29]. The Brunauer–Emmett–Teller (BET) surface area of the biochar used in this analysis was approx. 65 ± 5 m^2^∙g^−1^, which is a value similar to the results obtained by Aselkand et al. [54] and Tomczyk et al. [55] for the biochar from woodchips. This surface area is much larger than that of the biochar used by Klimek-Kopyra et al. [46] for water retention and nitrogen retention in soils (approx. 19 m^2^∙g^−1^) but significantly lower than that of the biochar applied in the sorption of heavy metals [56,57]. Moreover, the biochar used in our research was characterized by a large amount of surface functional groups (approx. 400 cmol∙kg^−1^).

The physical and chemical characteristics of the raw materials and input mixtures are presented in Malinowski [42]. According to data (Table 1), the DM of UFMSW was 56.8 wt% and it increased with an increase in the dose of the biochar added to the waste. These changes were not linear, mainly due to the heterogeneous material composition. The DM of individual mixtures was significantly lower than that in the study by Wolny-Koładka et al. [58], who reported DM equal to 64.5 wt%, but comparable to the values presented by Malinowski et al. [2]. Due to the low ash content in biochar, each addition of biochar to UFMSW decreased the ash content of the input mixtures. As in the case of DM, this relation was also not linear. The total C content (TC) before the application of biochar was close to the value reported by Wolny-Koładka et al. [58] (29.2 ± 5.2% DM) and slightly higher than that stated by Jędrczak and Suchowska-Kisielewicz [6] (23.7 ± 4.7% DM). Similarly to their results, in this research also, the addition of biochar increased the initial TC in the input mixtures. This trend was not observed for the total N content (TN). The TN was at a relatively high level, and the addition of biochar did not change this value (statistically non-significant differences). Moreover, Wolny-Koładka et al. [58] stated that TN in UFMSW is lower than that in the presented analysis—in their research, it amounted to 0.89 ± 0.19% DM, which also influenced the C/N ratio (over 30). In this research, C/N for UFMSW was relatively low (22.8); the addition of biochar increased this ratio, which is of great importance for the biological processing of waste. The differences between the initial ash content, DM, and TC for B0% and for the mixtures with high biochar doses (B10% and B20%) were statistically significant (*p*-value < 0.05).

### 3.2. Impact of Biochar Addition on the Biostabilization Process, Temperature Changes, and CO_2_ Concentration

In each analyzed run (with and without the addition of biochar), thermophilic temperatures (temperatures above 45 °C) were achieved (Figure 1). The course of temperature changes was similar to the results presented by Yuan et al. [51], Tom et al. [52], and Malinowski et al. [2], who used various types of bulking agents in the process of biostabilization or biodrying of MSW under laboratory conditions. The lowest air-flow rate (0.1) resulted in the maximum temperatures not reaching the level of 65 °C, which may indicate a lower enzymatic activity of microorganisms as a result of ineffective aeration of waste in the bioreactor [42]. The maximum process temperatures were reached the fastest with the highest air-flow rate value (0.4), on the very first day of the process. Usually, after the 12th day of the analysis, the temperature of the waste dropped to approx. 20 °C, which indicates a slowdown in microbiological processes. Therefore, Figure 1 shows the temperature course for the first 15 days of the process. Analogously, Figure 2, Figure 3 and Figure 4, presenting the concentrations of CO_2_ and NH_3_ emission, also present data for the same period of time.

Steiner et al. [59], Jindo et al. [60], and Malińska et al. [40] found that while composting waste with biochar addition, the recorded temperatures were significantly higher in relation to the control (without biochar addition). Contrary to the composting process, for the aerobic biostabilization of UFMSW, higher temperatures compared to the control on the addition of biochar were not noted.

The thermophilic phase lasted in most cases for 3–4 days, which is typical of experiments conducted with UFMSW under laboratory conditions [1,2,51,52]. It was found that the addition of 3% and 5% of biochar significantly extends the thermophilic phase and shifts the maximum temperature of the process by at least 1 day. Temperatures above 45 °C were maintained for at least 6 and 8 days for B3% and B5% runs, respectively. In the conducted aerobic biostabilization processes, there were no external sources of heat used, so the temperature increase observed for B3% and B5% resulted, among others, from increased activity of microorganisms. Biochar improves aeration (due to its high nano-porosity and large surface area) [61] and also increases the activity of microorganisms in the composting process [60]. Contreras-Cisneros et al. [62] and Wei et al. [63] have reported that the positive effect of biochar on the biological treatment of waste is also related to the ability of the biochar to bind water molecules.

Malinowski [42] found that with an increase in the biochar dose applied to UFMSW, the AFP of the input mixtures also increases, which should result in greater oxygen availability for microorganisms. However, in the case of aerobic biostabilization with a low air-flow rate (0.1) and the addition of a high amount of biochar (10% and 20%), the presence of anaerobic zones in the waste mixtures and a lack of oxygen in the exhaust gases were noted between the 2nd and 6th day of the process [42]. At higher air-flow rates (0.2 and 0.4), Malinowski [42] stated that the oxygen concentration in the exhaust gases is the lowest on the addition of 10% and 20% of biochar.

Table 2 shows the characteristics of UFMSW after 21 days of the biostabilization process with the addition of biochar at three different air-flow rates. Changes in the parameters of non-supplemented UFMSW (B0%), especially in TC and TN, were typical for the biostabilization of this type of waste [2].

After the process, the DM increased, and the higher the air-flow rate, the greater the increase in the DM content in relation to the initial value was observed. It is important that in the runs with a low addition of biochar (less than 3%), there was a much higher water loss than in the case of B5%, B10%, and B20%. Differences in water loss in the two groups mentioned (addition of less than 3% and more than 5% of biochar) are statistically significant at the *p*-value < 0.05. This confirms the assumption that even in the process of UFMSW biostabilization, the addition of biochar contributes to water accumulation in the processed waste, as was observed in the case of composting of green waste with manure or sewage sludge [28,33,60].

Based on the conducted research, it was found that the addition of biochar at a dose of 3% and higher contributes to a reduction in N_loss_ (Table 2). The average N_loss_ value in the case of B0% and B1.5% runs was 34.5 ± 3.6%, while for B3%, B5%, B10%, and B20%, it was only 12.4 ± 3.5%. The lowest N_loss_ values were recorded in the run with the addition of 20% of biochar.

The correct course of the UFMSW biostabilization process is also confirmed by the decrease in the TC content in the waste after a 3-week treatment. The highest TC losses (about 30% in relation to the initial content) were observed for B5% in the case of air-flow rates of 0.2 and 0.4. For B10% and B20%, the losses in TC after 21 days of the process exceeded 20%. It is caused, among others, by the extension of the thermophilic phase in these runs and/or the intensification of the microbial activity. In none of the performed runs was the required TC content after the process (i.e., total organic C below 20% DM [64]) obtained (the content of total organic C was not tested; however, for MSW, this value is statistically similar to TC [65]). This observation indicates that the processing of UFMSW in the maturation phase is needed. Similar conclusions, despite the application of various bulking agents or conducting UFMSW biostabilization without additives, were noted in the research papers by Jędrczak and Suchowska-Kisielewicz [6], Połomka and Jędrczak [64], and Vaverková et al. [66].

Figure 2 shows the averaged changes in the CO_2_ concentration in the exhaust gases directed from the bioreactor to the biofilter. As in the studies by Malinowski et al. [2] and Beck-Friis et al. [67], the dynamics of CO_2_ concentration release in the bioreactor were highly influenced by the temperature regime. A sharp increase in the CO_2_ concentration in the process gases was recorded for air-flow rates of 0.1 and 0.2 in the first days of the process, between the 1st and 6th day. At the lowest air flow (0.1), the highest values of CO_2_ concentration, exceeding 18%, were found for B10% and B20%. The high CO_2_ content in these cases is associated with the intensification of the process and a significant decrease in the oxygen concentration in the exhaust gases [42]. At higher air-flow rates, the concentration of CO_2_ was the highest for B3% and B5%. For tests with an air-flow rate of 0.4, the concentration of CO_2_ in the exhaust gases did not exceed 5% in any run.

Biochar additives can both reduce and increase CO_2_ concentration in the exhaust gases from UFMSW biostabilization. A method that allows for a reduction in the CO_2_ concentration, and, consequently, the CO_2_ emission into the atmosphere is the use of CaO, which inhibits the growth of microorganisms and therefore also the intensity of the process. Such an effect of CaO application in biological waste treatment processes was confirmed by Wolny-Koładka et al. [58].

### 3.3. Impact of Biochar Addition on NH_3_ Emission

Addition of biochar significantly decreases the concentration of NH_3_ in the exhaust gases (Figure 3). In the case of 0.2 and 0.4 aeration, a 1.5% share of biochar lowers NH_3_ emission; biochar doses of 3% and 5% were found to be effective in limiting NH_3_ emission for all tested aeration flows; whereas for B10% and B20%, practically no NH_3_ was detected in the outlet.

For all tested air-flow rates, a strong correlation between the content of biochar in the mixtures and accumulated NH_3_ emission was observed (Figure 4). This indicates that with an increase in the share of biochar addition, the accumulated NH_3_ emission decreases. For the addition of 1.5% of biochar, in the case of two tested air flows, 0.2 and 0.4, the gaseous emission of NH_3_ was reduced within the period of 15 days by 40% and 15%, respectively. For higher biochar doses, the reduction was even more significant and occurred for all the tested air flows: for B3%, it ranged from 93.6% (air flow 0.4) to 98.0% (air flow 0.2); for B5%, it ranged from 97.5% (air flow 0.4) to nearly 100% (air flow 0.1); and for B10% and B20%, practically no NH_3_ emission was noted.

Changes in the aeration rate influence NH_3_ concentration and emission, which is observed mainly for B0% and B1.5% since higher biochar shares substantially limit NH_3_ in the exhaust gases. Lower air-flow rates result in higher values of NH_3_ concertation; however, for all the tested air-flow, the tendency for NH_3_ concentration changes over time was similar—the peaks were noticed in the first days of the process and corresponded to the temperature peaks. In the case of control runs (B0%), the accumulated NH_3_ emission on the 15th day was at a similar level for 0.1 and 0.2 air-flow rates; the highest aeration rate (0.4) resulted in an increased NH_3_ emission (up to 1.7 g NH_3_·(kg DM_init_)^−1^), which proves that higher aeration rates favor the release of gaseous NH_3_ during the process. The same observation was noted by De Guardia et al. [19], who researched the composting of wastewater sludge mixed with woodchips in a series of tests with diversified air-flow rates, and by Puyuelo et al. [23], who tested aeration modes in the composting of organic fraction of MSW supplemented with pruning waste as a bulking agent.

The accumulated NH_3_ emission in the case of the reference mixture (B0%), which varied from approx. 1.2 to 1.7 g NH_3_·(kg DM_init_)^−1^ depending on the air-flow rate, was at a similar level as in the research by Beck-Friis et al. [67], who composted mixtures of wheat straw and source-separated household organics, reaching the cumulative emission rates between approx. 2.8 and 3.3 g NH_3_·(kg DM_init_)^−1^ after 15 days, or by Eklind et al. [12], who investigated the composting of household waste mixed with straw in different process conditions, determining the accumulated emission in the range from 0.9 to as much as 5.1 g NH_3_·(kg DM_init_)^−1^ within 15 days of the process. Eklind et al. [12] found that higher process temperatures result in a significant increase in NH_3_ emission. The addition of biochar seems to mitigate this effect, since in the case of B3% and B5%, for which temperatures exceeded 45 °C for a couple of days (at aeration 0.2 and 0.4), the NH_3_ emission not only did not rise but also was much lower than for the control runs (B0%), characterized by a shorter period of thermophilic temperatures.

The similar positive influence of biochar addition on lowering NH_3_ emission was observed by Janczak et al. [27], who researched aerobic treatment (composting) of poultry manure and wheat straw mixtures. The accumulated NH_3_ emission measured by them reached approx. 5.5 g NH_3_·(kg DM_init_)^−1^ in the non-supplemented process and was reduced by 30% and 44% for mixtures containing 5% and 10% of biochar, respectively.

## 4. Conclusions

The conducted research showed that biochar addition to UFMSW in a small dose has a positive effect on the biostabilization process of UFMSW. The temperature changes in the processed waste during biostabilization were typical for this process; however, it was found that the addition of 3% and 5% of biochar contributed to the extended duration of the thermophilic phase. Changes in the CO_2_ concentration in the exhaust gases corresponded to the course of temperature changes. The highest concentration of CO_2_ was recorded at the lowest air-flow rate (0.1) on the addition of 10% and 20% of biochar. The addition of biochar in the amount of 5% and more reduces the water loss in the processed waste and also has a positive effect on the reduction of the C content, especially at air-low rates of 0.2 and 0.4. Moreover, the studies proved that the addition of biochar in the amounts of 3% and higher causes N retention in the waste, which results in a significant reduction in the NH_3_ emission from the processed waste compared to the control (UFMSW biostabilization process without the addition of biochar). The tests showed that at all the applied air-flow rates, the NH_3_ emission in the exhaust gases from the bioreactors was reduced by over 90% for B3% and B5% and in the case of the addition of 10% and 20% of biochar, practically no NH_3_ emission was noted.

## Figures and Tables

**Figure 1 materials-15-01771-f001:**
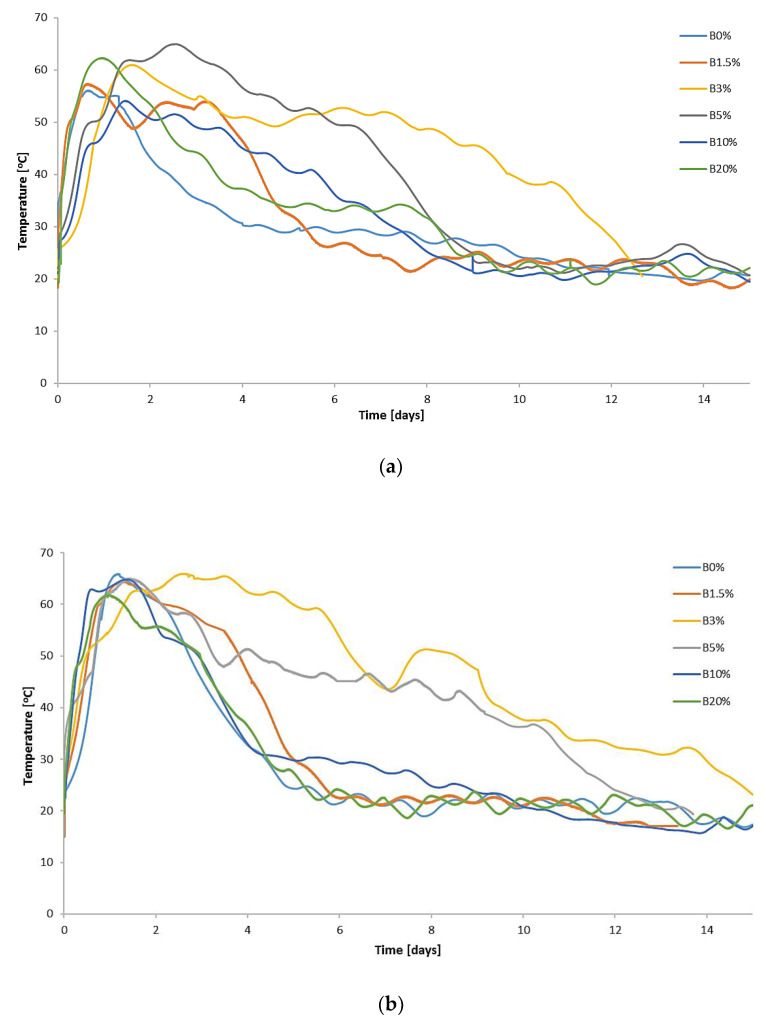
Temperature changes in the bioreactor during aerobic biostabilization: (**a**) air-flow rate of 0.1; (**b**) air-flow rate of 0.2; (**c**) air-flow rate of 0.4 m^3^·d^−1^·(kg organic DM)^−1^.

**Figure 2 materials-15-01771-f002:**
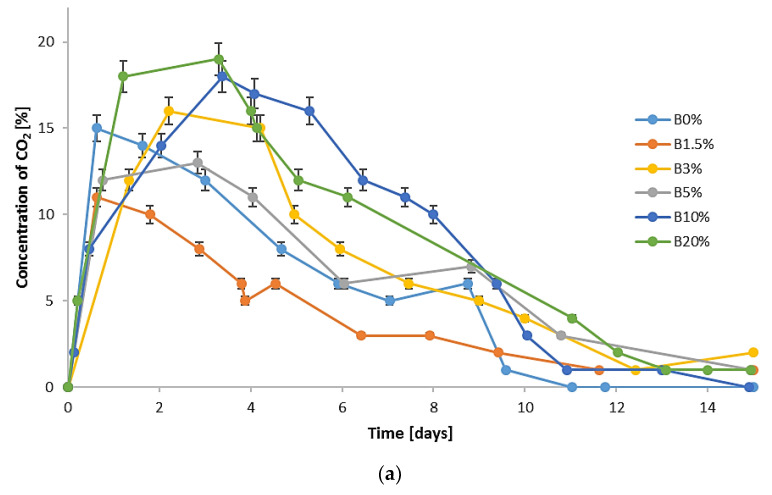
CO_2_ concentration changes in the exhaust gases during aerobic biostabilization: (**a**) air-flow rate of 0.1; (**b**) air-flow rate of 0.2; (**c**) air-flow rate of 0.4 m^3^·d^−1^·(kg organic DM)^−1^.

**Figure 3 materials-15-01771-f003:**
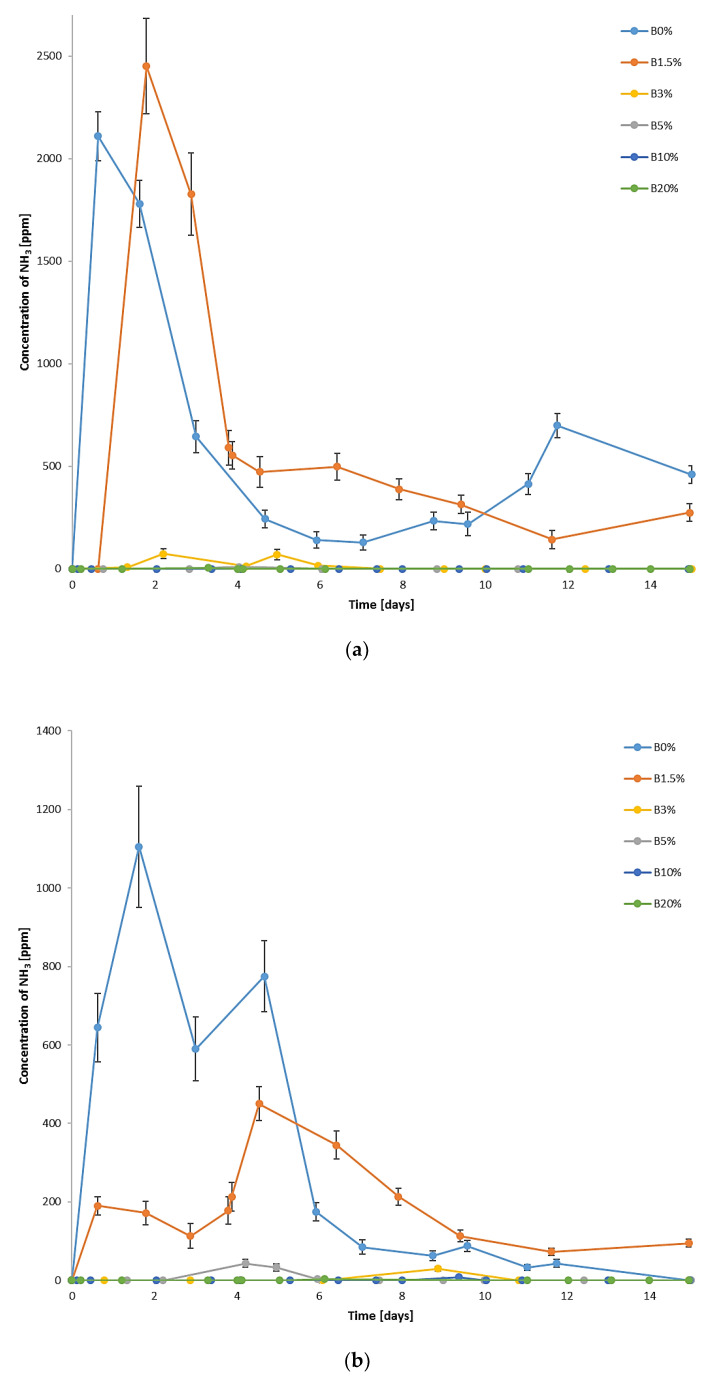
NH_3_ concentration changes in the exhaust gases during aerobic biostabilization: (**a**) air-flow rate of 0.1; (**b**) air-flow rate of 0.2; (**c**) air-flow rate of 0.4 m^3^·d^−1^·(kg organic DM)^−1^.

**Figure 4 materials-15-01771-f004:**
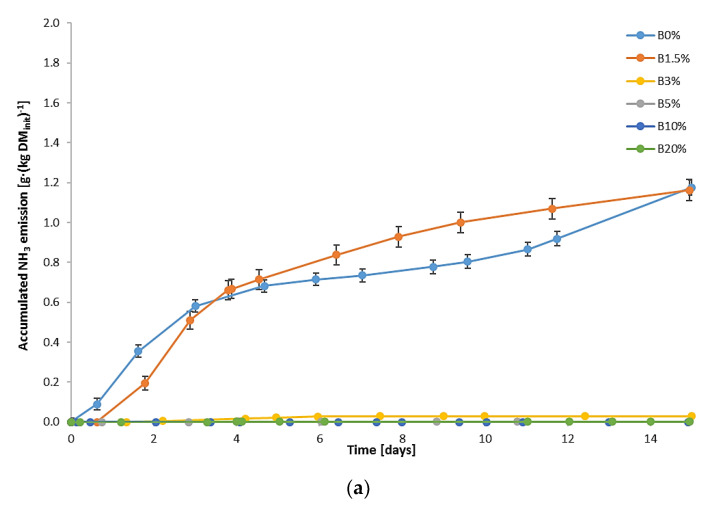
Accumulated NH_3_ emission in the exhaust gases during aerobic biostabilization: (**a**) air-flow rate of 0.1; (**b**) air-flow rate of 0.2; (**c**) air-flow rate of 0.4 m^3^·d^−1^·(kg organic DM)^−1^.

**Table 1 materials-15-01771-t001:** Selected physicochemical properties of mixtures without biochar (B0%) and with biochar at five different doses (run 2, 3, 4, 5, and 6).

Parameters	Unit	Run 1 (B0%)	Run 2 (B1.5%)	Run 3 (B3%)	Run 4 (B5%)	Run 5 (B10%)	Run 6 (B20%)
DM	wt%	56.8 ± 0.9	53.6 ± 4.4	60.8 ± 4.1	57.4 ± 2.0	61.9 ± 2.1	63.7 ± 2.4
Ash content	% DM	52.1 ± 0.8	52.9 ± 0.7	51.4 ± 1.8	50.5 ± 3.6	44.7 ± 6.3	39.0 ± 4.4
TC	% DM	28.2 ± 1.4	27.7 ± 2.4	27.0 ± 3.1	30.9 ± 4.2	34.6 ± 2.3	40.6 ± 3.2
TN	% DM	1.24 ± 0.11	1.23 ± 0.12	1.31 ± 0.16	1.26 ± 0.09	1.30 ± 0.17	1.23 ± 0.14
C/N	-	22.8	22.5	20.6	24.6	26.6	32.9

Mean ± standard deviation of the mean (*n* = 3). DM—dry matter; TC—total carbon; TN—total nitrogen.

**Table 2 materials-15-01771-t002:** Characteristics of UFMSW and biochar mixtures after 21 days of the aerobic biostabilization process.

Parameters	Unit	Run 1 (B0%)	Run 2 (B1.5%)	Run 3 (B3%)	Run 4 (B5%)	Run 5 (B10%)	Run 6 (B20%)
Air-flow rate: 0.1 m^3^·d^−1^·(kg organic DM)^−1^							
DM	wt%	65.3 ± 2.6	64.1 ± 7.4	70.2 ± 2.9	63.1 ± 1.9	68.1 ± 1.6	68.5 ± 3.0
Water loss	%	19.7	22.7	23.8	13.6	16.6	13.2
Ash content	% DM	59.5 ± 1.8	61.6 ± 2.2	59.6 ± 2.7	58.9 ± 4.2	56.9 ± 2.0	46.6 ± 3.4
TC	% DM	22.5 ± 1.3	24.0 ± 1.4	22.4 ± 4.2	25.7 ± 0.9	28.7 ± 2.0	36.4 ± 2.7
TN	% DM	0.87 ± 0.12	0.97 ± 0.18	1.25 ± 0.28	1.32 ± 0.11	1.41 ± 0.12	1.39 ± 0.22
C/N	-	25.8	24.7	18.0	19.4	20.4	26.2
N_loss_	%	38.4	32.4	18.2	10.1	15.1	5.8
Air-flow rate: 0.2 m^3^·d^−1^·(kg organic DM)^−1^							
DM	wt%	74.0 ± 2.9	70.0 ± 5.6	74.1 ± 3.0	66.7 ± 3.0	70.1 ± 1.3	70.9 ± 2.4
Water loss	%	39.8	35.4	33.9	21.8	21.7	19.9
Ash content	% DM	59.1 ± 1.8	59.5 ± 1.7	60.1 ± 1.7	63.9 ± 1.7	57.0 ± 2.1	51.4 ± 2.7
TC	% DM	22.7 ± 1.6	23.8 ± 1.4	22.2 ± 2.9	22.6 ± 2.2	26.9 ± 1.9	31.6 ± 2.6
TN	% DM	0.89 ± 0.10	0.98 ± 0.13	1.29 ± 0.11	1.42 ± 0.19	1.45 ± 0.22	1.42 ± 0.23
C/N	-	25.4	24.6	17.2	16.0	18.6	22.2
N_loss_	%	36.4	29.5	16.3	11.4	12.7	12.5
Air-flow rate: 0.4 m^3^·d^−1^·(kg organic DM)^−1^							
DM	wt%	76.5 ± 2.4	72.8 ± 6.9	74.8 ± 3.8	68.1 ± 1.9	70.7 ± 2.2	72.1 ± 2.6
Water loss	%	45.6	41.4	35.5	25.1	23.3	23.0
Ash content	% DM	60.5 ± 1.9	61.6 ± 1.7	60.5 ± 1.8	61.6 ± 2.5	56.5 ± 3.1	51.4 ± 1.6
TC	% DM	21.9 ± 1.1	22.6 ± 1.2	21.9 ± 2.8	21.4 ± 1.1	27.1 ± 1.8	30.6 ± 2.1
TN	% DM	0.89 ± 0.06	0.97 ± 0.13	1.30 ± 0.12	1.36 ± 0.10	1.48 ± 0.27	1.47 ± 0.23
C/N	-	24.5	23.3	16.8	15.7	18.4	20.7
N_loss_	%	37.9	32.5	15.9	11.4	10.1	9.3

Mean ± standard deviation of the mean (*n* = 3). DM—dry matter; TC—total carbon; TN—total nitrogen.

## Data Availability

All data derived during the experiments are available at the Faculty of Production and Power Engineering, University of Agriculture in Krakow.

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
