# Peer review of "Impact of Biochar Addition and Air-Flow Rate on Ammonia and Carbon Dioxide Concentration in the Emitted Gases from Aerobic Biostabilization of Waste"

_materials, 2022, doi:10.3390/ma15051771_

Round 1

Reviewer 1 Report

In this paper, the effects of biochar addition and air circulation on ammonia emission and carbon dioxide concentration in exhaust gas during the biostabilization process of municipal solid waste components were studied. The paper shows a great way to reduce greenhouse gas (NH3 and CO2) emissions. Thus, I recommend the publication of this paper in Materials after minor revision. Some questions and suggestions are listed as follows.

1. All the pictures in this paper should be redone to make them clearer.

2. It is recommended to increase the BET test of biochar, and use data to confirm biochar improves aeration (due to its high nano-porosity and large surface area).

3. The method of N fixation by the addition of biochar should be verified experimentally to explain the mechanism of action of biochar.

4. The surface functional group of biochar should be characterized by FT-IR test to explore the binding mode of water.

5. Many studies on the adsorption of organic matter by biomass materials have been published. For a better understanding of mechanistic studies, authors are encouraged to cite the following articles:

(i): Sulfhydryl-modified sodium alginate film for lead-ion adsorption.

 https://doi.org/ 10.1016/j.matlet.2019.07.055

(ii): Effects of biochar on soil available inorganic nitrogen: a review and meta-analysis.

https://doi.org/ 10.1016/j.geoderma.2016.11.004

Author Response

Response to Reviewer 1 Comments

We would like to thank the Reviewer 1 for kind evaluation of the paper and the positive feedback of our work. The detailed response are presented in the attached file.

Reviewer 2 Report

Dear Authors,

Herein, I submit my comments for initial submission of the manuscript entitled: “Impact of biochar addition and air-flow rate on ammonia emission and carbon dioxide concentration in the emitted gases during biostabilization of undersized fraction from municipal solid waste.”

Climate change affects fauna, flora, people's lives. Today, there is no doubt that these changes are caused by human activity. Air pollution seems to be the main factor. Improving waste treatment is one of the key factors in slowing down climate change. This manuscript focuses on reducing CO2 and NH3 emissions by adding biochar before the "undersized fraction from municipal solid waste" treatment process. The addition of biochar causes an increase in CO2 emissions but, on the other hand, a dramatic decrease in NH3 emissions also due to the extension of the thermophilic phase. The presented results seem to unconditionally contribute to the reduction of nitrogen in the gas outlet. In reverse, increasing carbon concentration is detrimental to the addition of biochar to UFMSW.

I consider this manuscript suitable for publication. However, some ambiguities arose during the reading. Therefore, I came to a conclusion to encourage you to review it again. Please, use my comments listed below.

COMMENTS:

The whole paper:

  1. If possible, please shorten the paper title.
  2. Please indicate in the manuscript whether it is possible to include an element in the process that would reduce the carbon dioxide content.

Chapter 3.1:

  1. Is the water retention in biochar main mechanism of the biostabilization? Have you considered other materials that are able to retain water (with the same or better effect as the biochar)?
  2. How is the rest of the material left in the bioreactor handled? Doesn't this represent a greater environmental burden in terms of nitrogen and carbon content than the gas discharges themselves?

Thank you,

Reviewer.

Author Response

Response to Reviewer 2 Comments

We would like to thank the Reviewer for kind evaluation of the paper and the positive feedback of our work. The detailed response are presented in the attached file.

Reviewer 3 Report

The submitted manuscript describes the effect of adding various biochar doses to the biostabilization process of MSW and the related gaseous emissions. This biochar application is less studied compared to soil applications or as adsorbent for wastewater treatment. The Introduction section is  well-written and provides an overview of this field. The experimental part is clear and can be reproduced by other researchers. The results are statistically sound and compared adequately to earlier works. I recommend publication of the manuscript after the following corrections are made: 

  1. line 147 - it is not clear what the components of the separated biodegradable waste are
  2. line 150 - the residence time for pyrolysis should be mentioned
  3. line 155 - correct to 'the time of analysis'
  4. line 160 - what is meant by 'wet weight' ? what was the moisture in biochar ?
  5. Figures 1, 2, 3, 4 - why the authors did not monitor the temperature changes and other parameters for 21 days ?
  6. There are some syntax errors and grammar mistakes throughout the manuscript. An English language revision is recommended.

Author Response

Response to Reviewer 3 Comments

We would like to thank the Reviewer for the kind review and accurate suggestions. We did our best to revise the manuscript according to Reviewer’s remarks. The detailed responses are presented in the attached file.

Reviewer 4 Report

The manuscript  deals with the "effects of biochar and air flow rate on ammonia and CO2 emissions from municipal wastes during biostabilization process".

1. Page 1, Line 12; "Application of additives to waste may influence the course of this process ..." Which process?!

2. Page 1, Line 13; "..gases and ammonia (NH3) emission from the process."" Same comment as above!

3. Page 1, Line 21; "The highest concentration of CO2..." Mention the values.

4. Page 4, Line 159; "Six different doses of biochar were added..." What kinds of biochar? Biochar extracted from?

5. Mention the size of biochar used in the study.

6. If applicable, the BET analysis of biochar should be mentioned in the manuscript.

7. Quality of Figures 1, 2 and 3 should be improved.

Author Response

Response to Reviewer 4 Comments

We would like to thank the Reviewer for the very detailed and accurate review. We did our best to revise the manuscript according to Reviewer’s suggestion. The detailed responses are presented in the attached file.

Round 2

Reviewer 4 Report

Reviewers' comments have been addressed.